# Rapid response of habitat structure and aboveground carbon storage to altered fire regimes in tropical savanna

Shaun R. Levick[1,2,3], Anna E. Richards[2], Garry D. Cook[2], Jon Schatz[2], Marcus Guderle[1], Richard J. Williams[2], Parash Subedi[3], Susan E. Trumbore[1], and Alan N. Andersen[3]

[1]Max Planck Institute for Biogeochemistry, Hans-Knoell-Str. 10, 07745 Jena, Germany
[2]CSIRO Land and Water, PMB 44, Winnellie, 0822 NT, Australia
[3]Research Institute for the Environment and Livelihoods, Charles Darwin University, NT 0909, Australia
**Correspondence:** Shaun R. Levick (shaun.levick@csiro.au)

**Abstract.** Fire regimes across the globe have been altered through changes in land-use, land management and climate conditions. Understanding how these modified fire regimes impact vegetation structure and dynamics is essential for informed biodiversity conservation and carbon management in savanna ecosystems. We used a fire experiment at the Territory Wildlife Park (TWP), northern Australia, to investigate the consequences of altered fire regimes for vertical habitat structure and aboveground carbon storage. We mapped vegetation three-dimensional (3D) structure in high spatial resolution with airborne LiDAR, across 18 replicated 1 ha plots of varying fire frequency and season treatments. We used LiDAR-derived canopy height and cover metrics to extrapolate field-based measures of woody biomass to the full extent of the experimental site ($R^2 = 0.82$, RMSE = 7.35 t C ha$^{-1}$), and analysed differences in aboveground carbon storage and canopy structure among treatments. Woody canopy cover and biomass were highest in the absence of fire (76 % and 39.8 t C ha$^{-1}$) and lowest in plots burnt late in the dry season on a biennial basis (42 % and 18.2 t C ha$^{-1}$). Woody canopy vertical profiles differed among all six fire treatments, with greatest divergence in height classes < 5 m. The magnitude of fire effects on vegetation structure varied along the environmental gradient underpinning the experiment, with less reduction in biomass in plots with deeper soils. Our results highlight the large extent to which fire management can shape woody structural patterns in savanna landscapes, even over time frames as short as a decade. The structural profile changes shown here, and the quantification of carbon reduction under late dry season burning, have important implications for habitat conservation, carbon sequestration, and emission reduction initiatives in the region.

## 1 Introduction

Fire is an integral component of the functioning of savanna ecosystems, exerting top-down control on woody vegetation structure (Bond and Keeley, 2005; Sankaran et al., 2005). Savanna fires restrict vegetation vertical growth through a "fire-trap" mechanism, whereby young trees are constrained to low woody resprouts under high fire frequencies (Higgins et al., 2000; Freeman et al., 2017). A lengthening of the fire-free interval allows trapped woody plants to grow above flame height, enabling them to reach mid- and upper canopy heights, with long-term consequences for size-class distribution and structural heterogeneity (Helm and Witkowski, 2012; Levick et al., 2015a).

Three-dimensional (3D) heterogeneity of vegetation has long been valued as a key factor promoting faunal diversity through increased niche diversity and availability (MacArthur and MacArthur, 1961; MacArthur, 1964). The structural modifications that fires impart on savanna vegetation have been shown to impact both vertebrate (Woinarski et al., 2004) and invertebrate (Andersen et al., 2012) taxa. Fire-driven structural changes to savanna vegetation also have important implications for climate regulation, as savanna fires contribute significantly to atmospheric emissions of greenhouse gases through biomass combustion (Hurst et al., 1994; van der Werf et al., 2010). Despite the importance of quantifying fire induced changes to 3D structure in savanna vegetation, current understanding of magnitudes and spatial patterns remains limited, and savanna fires represent large uncertainty in global vegetation models (Higgins et al., 2007; Scheiter et al., 2013). Gaining better understanding of how different fire regimes impact savanna vegetation structure is becoming increasingly urgent in the face of changing climate and land-management conditions that are triggering variations in the timing, frequency, intensity and duration of fires in the tropical biome (Alencar et al., 2015).

Fire frequency in Australian savannas is particularly high, with many regions burning twice in every three years on average (Beringer et al., 2014). Many of these fires occur late in the dry season, producing high intensity burns that result in simplified vegetation structure (Bowman et al., 1988; Lehmann et al., 2009; Ondei et al., 2017). There are widespread concerns that such fire regimes are linked to dramatic declines in faunal populations, through the removal of ground layer vegetation (Lawes et al., 2015; Legge et al., 2015; Woinarski et al., 2015). Methane and nitrous oxide emissions from savanna fires are included in Australia's national greenhouse-gas accounts, and are responsible for approximately 3 % of total accountable greenhouse-gas emissions (Meyer et al., 2012). There is considerable interest in reducing the frequency and intensity of fires in northern Australia through strategic early dry season (April to July) burning, in order to reduce both greenhouse gas emissions and certain components of biodiversity decline (Russell-Smith et al., 2013). As such, the Australian Government has implemented legislation enabling landowners to claim carbon credits for reducing greenhouse gas emissions from savanna fires through early dry season burning (Carbon Farming Initiative - Emissions Abatement through Savanna Fire Management Methodology Determination 2015, Department of Environment and Energy). Such changes to fire regimes in northern Australia are also likely to increase carbon sequestration in the landscape (Murphy et al., 2010; Richards et al., 2012), although there is currently no approved methodology for incorporating this into the national accounts. While much attention is currently being given to reducing the extent and frequency of late season fires in northern Australia, it is important to recognise that savannas have evolved with fire (Bond and Keeley, 2005; Durigan and Ratter, 2016) and excluding fire would be detrimental to certain savanna specialists that favour more open and grassy habitat. The challenge is finding the best mix of patches of different regimes across connected landscapes.

Understanding of how different fire regimes impact habitat structure and carbon dynamics in tropical savannas can be enhanced through detailed 3D measurements of vegetation structure at sites subject to long-term, replicated experimental fire treatments. Traditional field-based inventory techniques are limited in their ability to quantify 3D structure, but light-detection and ranging (LiDAR) can now achieve this with high accuracy and precision in a repeatable and transferable manner (Lefsky et al., 2002; Levick and Rogers, 2008). Airborne LiDAR has a proven record in providing detailed 3D representations of

savanna vegetation structure across time and space (Smit et al., 2010; Levick et al., 2012, 2015b; Goldbergs et al., 2018), but has yet to be used for assessing vegetation biomass and structural diversity responses to experimental fires in savannas.

Northern Australia has a long history of savanna fire experiments (Williams et al., 2003), including the ongoing 'Burning for Biodiversity' experiment at the Territory Wildlife Park that has applied six fire treatments in three replicated blocks since 2004 (Scott et al., 2010). Here we integrate field-based measurements of vegetation structure with airborne LiDAR to determine how variation in fire frequency and season affects the 3D habitat structure and aboveground carbon storage of woody vegetation. Our specific aims are to: i) explore how vegetation carbon storage and structural diversity respond to increasing fire frequency; and ii) quantify the structural impact of late-season fires compared to early-season fires. We use airborne LiDAR data to provide greater spatial coverage than can be achieved with field sampling alone, and to gain better understanding of how reliably LiDAR could be used to assess savanna carbon dynamics in instances where field data may not be available or attainable.

## 2   Methods

### 2.1   Study site and experimental design

The Territory Wildlife Park is located 40 km south of Darwin in Australia's Northern Territory (Figure 1). The vegetation at the site is a mixed open forest and woodland savanna dominated by *Eucalyptus miniata* A.Cunn. ex Shauer, *Eucalyptus tetrodonta* F.Muell. and *Corymbia bleeseri* (Blakely) K.D.Hill and L.A.S. Johnson, with a grassy understory dominated by *Pseudopogonatherum contortum* (Brongn.) A.Camus, *Sarga intrans* F.Muell. ex Benth. and *Eriachne triseta* Nees ex Steud (Scott et al., 2010) . The soils are relatively shallow (0.5 to 1 m deep) gravelly red earths (Petroferric Red Kandosol) (Isbell, 2002) of the Kay land system within the Koolpinyah land surface group, and have developed predominantly from deeply weathered sandstones, siltstones and shales (Wood et al., 1985). The climate is wet-dry tropical with greater than 90 % of annual rainfall (mean 1401 mm) falling in the wet season from November to April, and mean monthly maximum and minimum temperatures between 33.1 °C and 20.9 °C (Bureau of Meteorology, Commonwealth of Australia).

The fire experiment consists of 18 1-ha plots grouped into 3 blocks (A, B, C) arranged along a north-south transect (Figure 1). Soils are deeper at the southern end, and the C block has higher soil moisture given its proximity to a small drainage line. Six fire treatments were randomly assigned to each block at the start of the experiment: unburnt plots (U); plots burnt at fire return intervals of 1 (E1), 2 (E2), 3 (E3) and 5 (E5) years in the early dry season (E2); and plots burnt every 2 years (L2) in the late dry season (Table 1). Prior to implementation of the burning treatments in 2004, all areas had been unburnt for at least 14 years when fire records started (except for a fire in 1992 and again in 2000 in the A block only).

During each experimental burn, fire intensity was estimated using the established relationship between rate of spread and fuel load (Williams et al., 1998). Rate of fire spread was determined from thermocouples positioned 5 cm above the soil surface in the flaming combustion zone, linked to buried electronic stop watches. Six timers were used in each 1 ha plot, arranged in a series of equilateral triangles, with 10 m sides. The rate of fire spread was also determined by observers using stop watches, manually recording the time of arrival at the points where the electronic watches were positioned. All points were marked by

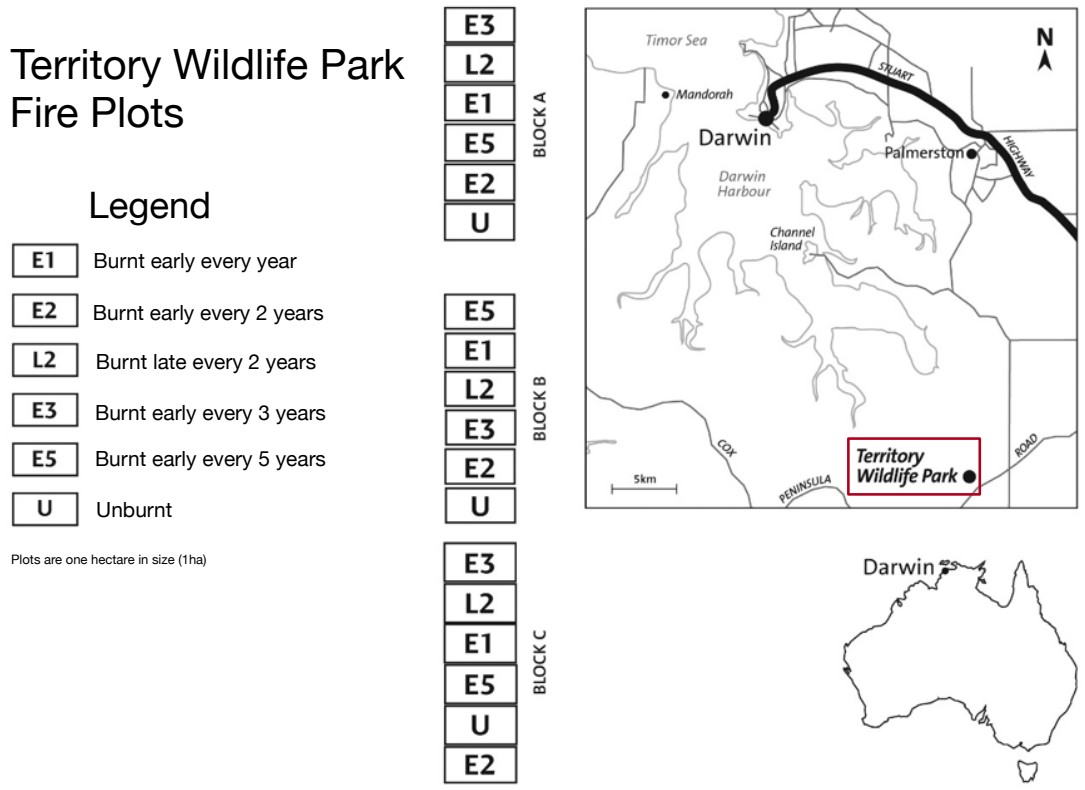

**Figure 1.** Location and experimental design of the Territory Wildlife Park fire manipulation experiment. Treatments were fist implemented in 2004. Soil depth and soil moisture increases from the northern to southern blocks.

**Table 1.** Fire regime characteristics of the Territory Wildlife Park experimental site. Data are for the period 2004-2013. Fire intensity values are the mean and standard error over the course of the experiment.

| Treatment | Season | Frequency (yrs) | Intensity (kW m$^{-1}$) | Times burnt |
|-----------|--------|-----------------|--------------------------|-------------|
| E1 | June | 1 | 589 $\pm$144 | 9 |
| E2 | June | 2 | 929 $\pm$20 | 4 |
| E3 | June | 3 | 424 $\pm$26 | 3 |
| E5 | June | 5 | 295 $\pm$69 | 2 |
| L2 | October | 2 | 1644 $\pm$131 | 5 |
| U | n/a | 0 | 0 | 0 |

star pickets and flagging tape. Fuel loads were determined prior to each fire by direct harvest and weighing. Ten replicate 0.5 m x 0.5 m fuel samples were cut for each plot. Fuel heat content was assumed to be 20 000 kJ per kg dry weight.

## 2.2   Field-based estimation of aboveground woody biomass

In each of the 18 plots, two 30 x 30 m subplots were established at the north-west and south-east corners, at least 10 m away from plot edges. In each subplot the species identity, location, height and diameter of all woody plants > 2 m in height was recorded. The location of each individual plant was recorded to 0.3 m accuracy using a differential GPS with post-processing (Trimble Inc.). Tree heights were recorded with a standard height pole (plants < 8 m) or clinometer (plants > 8 m), and stem diameter was recorded at 1.3 m with a diameter tape for all woody species except for the multi-stemmed shrubs *Calytrix exstipulata* and *Exocarpus latifolius*, in which case diameter was recorded at the stem base (0.1 m above the ground). Aboveground biomass was calculated for each individual tree using the equation developed by Williams et al. (2005):

$$lnABG = -2.0596 + 2.1561(lnD) + 0.1362(lnH)^2 \tag{1}$$

whereby AGB = aboveground biomass (kg), D = stem diameter (m), and H = tree height (m). Individual tree biomasses were then summed for each 30 X 30 m subplot. Estimated biomass values were converted to carbon terms on a per hectare basis assuming 50 % of biomass was carbon (t C ha$^{-1}$). This approach did not consider the contribution of small (< 2 m) multi-stemmed shrubs to the carbon pool.

## 2.3   Airborne LiDAR surveying and processing

We mapped 150 ha of the study area with airborne LiDAR in June 2013, 9 years after the start of the experiment. The airborne survey was conducted by Airborne Research Australia (ARA) with a full-waveform LiDAR sensor (RIEGL LMS-Q560) operated from a light fixed-wing aircraft (Diamond Aircraft ECO-Dimona). Flight-lines with > 50 % overlap were used to achieve double coverage of the plots (average flying height 300 m AGL, swath width 250 m, line spacing 125 m), and the RIEGL LMS-Q560 was operated at 240 kHz and 135 lines per second. Slow flying speed of less than 40 ms$^{-1}$ ensured high point densities along track, with an average return density of 22.28 m$^2$ and an average pulse spacing of 0.21 m.

Raw LiDAR data were processed with RiANALYZE (RIEGL Laser Measurement Systems GmbH) for decomposing the full waveforms into discrete returns. The ARA RASP open source software (RASP Version 0.98: manual, code and executables available from ARA on request) was used to orientate the point cloud to Cartesian coordinates and output the geolocated point cloud in the American Society for Photogrammetry and Remote Sensing (ASPRS) standard LAS format. All further point-cloud processing tasks were conducted with the LAStools suite of processing scripts (rapidlasso GmbH). The last returns were classified into ground and non-ground points for bare-earth extraction. A digital terrain model (DTM) was constructed from ground returns using a triangulated irregular network approach (TIN) at 0.25 m resolution. The DTM was used to normalize the *z* coordinate of vegetation returns to height above ground level (Figure 2).

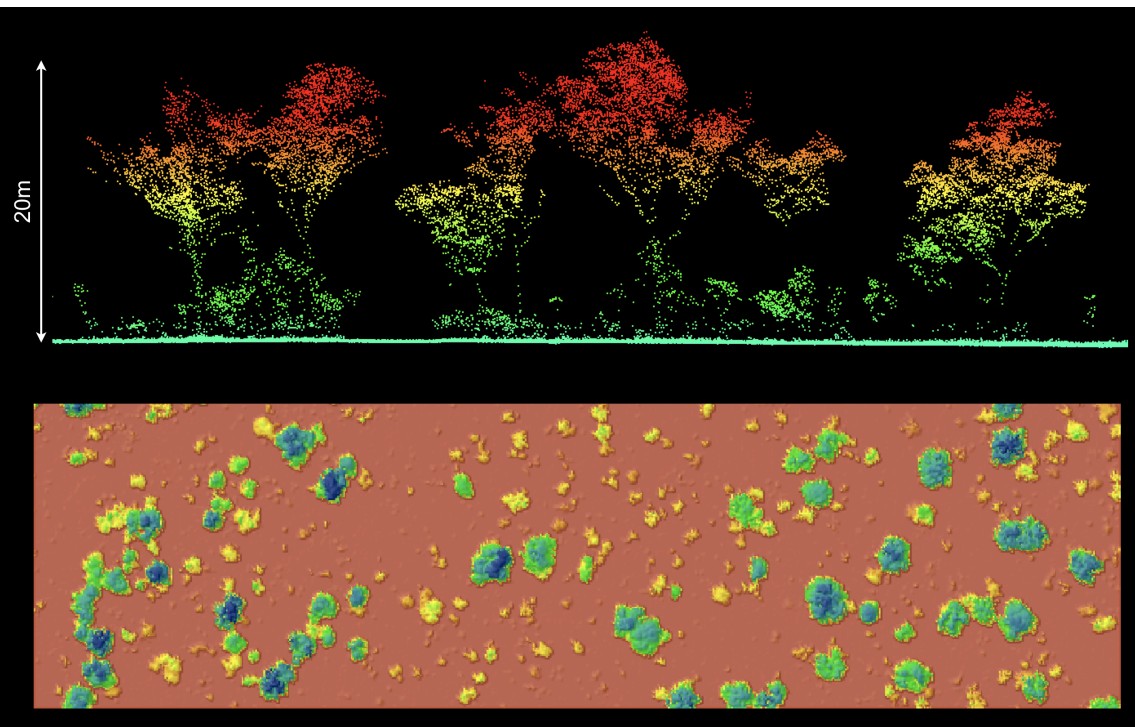

**Figure 2.** Cross-section through the normalized high resolution LiDAR point cloud (top) and aerial view of rasterised canopy height model (CHM) interpolation (bottom). The LiDAR point cloud provided excellent representation of both the vertical and horizontal structure of vegetation across the site.

## 2.4 Upscaling aboveground woody biomass estimates with airborne LiDAR

The normalized airborne LiDAR returns were clipped to the spatial extent of each field-measured 30 X 30 m subplot. Using the *lascanopy* tool within LAStools, we extracted a suite of 14 ecologically meaningful metrics describing vegetation structure from the point cloud: mean canopy height (MCH), quadratic mean canopy height (QMCH), canopy cover > 1 m (COV1),
5   canopy cover > 10 m (COV10), canopy density (DENS), kurtosis (KUR), skewness (SKE), standard deviation (SDE), canopy relief ratio (CRR), and a series of height quantiles (Q10, Q25, Q50, Q75, Q90). Using these 14 metrics as explanatory variables, we ran step-wise multiple linear regression with AIC minimization against the field-estimated biomass to identify the variables with the most explanatory power, and used them to construct a LiDAR-based biomass model. We applied the most robust model (in terms of explanatory power and RMSE) across the full extent of the airborne LiDAR coverage to examine the effects
10   of fire treatment on aboveground woody biomass.

## 2.5    Assessment of treatment effects

We digitally distributed six 30 m X 30 m subplots in each fire plot for statistical comparison of treatments effects. We used a linear mixed effect modelling approach, with Gaussian residual variance, to test the significance of fire treatment on woody canopy cover, canopy height and aboveground biomass. The models were implemented in R (R Core Team 2018) with the *nlme*
package (Pinheiro et al., 2018). Maximum likelihood (ML) was used to fit the models, with subplots included as a random effect nested within fire treatments. Fixed effects were fire treatment (E1, E2, E3, E4, E5, L2, U), block position (A, B, C) and the interaction between fire treatment and block position. Models were generated for all possible combinations of fixed effects, together with a null model consisting of only the random effects of the quadrat locations. Akaikie Information Criterion (AIC) scores for each of the models were compared to identify the most parsimonious model.

The impact of fire treatment on vegetation vertical profile distribution was explored my plotting the mean and 95 % confidence interval of LiDAR returns per 0.5 m height class. Statistical significance of treatment effects was tested pairwise on a per height class basis using a paired t-test.

## 3    Results

### 3.1    Estimation of aboveground woody biomass from airborne LiDAR

Airborne LiDAR proved valuable for upscaling woody biomass measurements from the field-plots to the full extent of the fire experiment (Figure 3). Only three woody canopy structural variables were retained in the step-wise linear regression procedure: mean canopy height (MCH), total canopy cover (Cov1m), and overstory canopy cover (Cov10m):

$$AGB = -6.524 + (-0.794 Cov10m) + (-0.345 Cov1m) + (14.881 MCH) \tag{2}$$

The distribution of model residuals showed no spatial trend nor relationship with the fire treatment. The degree of residual
error (RMSE = 7.35 t C ha$^{-1}$) provided acceptable confidence for inclusion of modelled biomass values in further analyses.

### 3.2    Effects of fire regime on woody canopy cover and aboveground biomass

Canopy cover decreased along the experimental gradient of fire frequency and season, ranging from about 75 % (SE = 1.7) in unburnt plots to 45 % (SE = 2.3) in late season bienniel plots (Figure 4a). These differences in canopy cover translated into similar patterns of biomass variation across the experiment (Figure 4b). The highest within-treatment variability for both cover
and biomass was found in the early season annual plots (E1).

The best model explaining variation in both woody cover and biomass was one in which fire treatment, block position, and the interaction between them was included (Table 2). Model performance was poorer when the interaction term was excluded ($\Delta$AIC = 48.91, 32.98 and 39.29 for woody cover, height and biomass respectively). When explanatory variables were considered independently, fire treatment was more influential than block position on variation in woody cover ($\Delta$AIC =

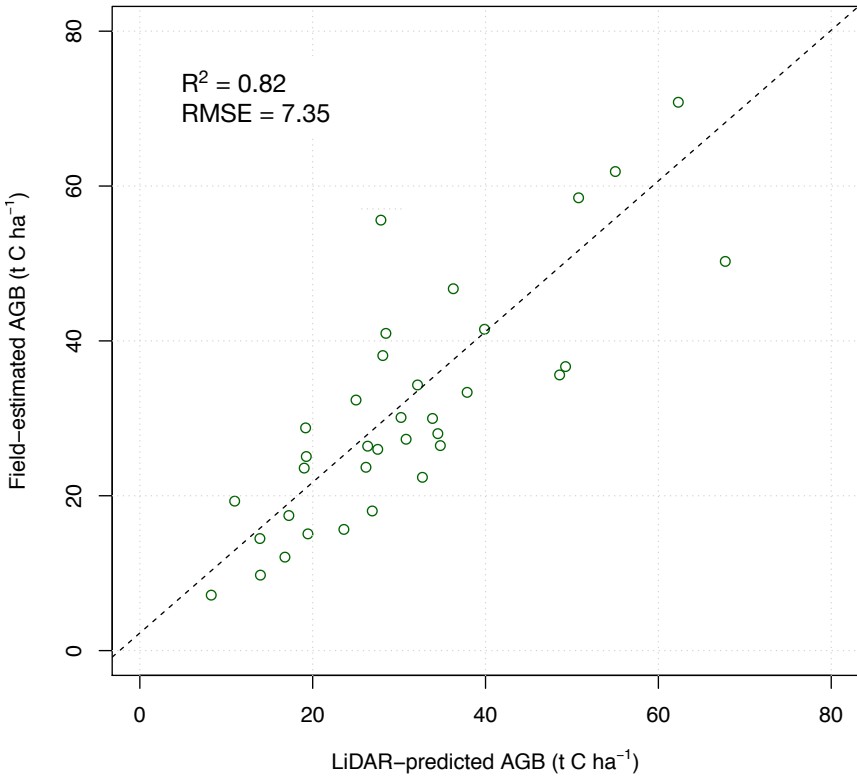

**Figure 3.** Relationship between field-estimated aboveground biomass and estimates predicted from airborne LiDAR metrics. Open green circles represent individual subplots (30 m X 30 m), dashed line shows the linear fit.

**Table 2.** Linear mixed model results of LiDAR estimated canopy cover, mean height and aboveground woody biomass

| Model terms | AIC *cover* | ΔAIC *cover* | AIC *height* | ΔAIC *height* | AIC *biomass* | ΔAIC *biomass* |
|---|---|---|---|---|---|---|
| Fire treatment * Block | 806.14 | 0.00 | 406.47 | 0.00 | 864.51 | 0.00 |
| Fire treatment + Block | 855.05 | 48.91 | 439.45 | 32.98 | 903.81 | 39.29 |
| Fire treatment | 876.59 | 70.45 | 466.43 | 59.96 | 937.87 | 73.34 |
| Block | 896.81 | 90.67 | 455.40 | 48.93 | 916.23 | 51.70 |
| Null model | 915.34 | 109.20 | 477.03 | 70.56 | 944.09 | 79.57 |

70.25 vs 90.67), but not for mean canopy height ($\Delta$AIC = 59.96 vs 48.93) or woody biomass ($\Delta$AIC = 73.34 vs 51.70). These results point to an important source of environmental variation arising from block position, which represents a gradient in soil depth and moisture availability across the experimental site.

When we consider the spectrum of increasing fire intensity occurring across the experimental treatments, we found that correlations between the reductions in aboveground biomass and fire intensity decreased along the soil depth and moisture availability gradient (Figure 5). In carbon terms, the early biennial fires on average caused a reduction of 10 t C ha$^{-1}$ compared to unburnt plots, whereas late biennial fires almost doubled that reduction to 19 t C ha$^{-1}$.

### 3.3 Fire effects on vertical habitat structure

In addition to the observed patterns in woody canopy cover and aboveground biomass, our LiDAR-based assessment also revealed substantial variation in canopy height profile distributions, derived from the number of LiDAR returns from different height levels (Figure 6). Most profiles were bimodal, with a peak at 1-2 m height and a smaller peak at 10-15 m. The clearest bimodal response was found in the early season triannual burns (Figure 6c), whereas early season annual and 5-yr burn profiles were more uniform (Figures 6a,d).

Keeping fire frequency constant (biennial) and exploring the effects of fire season highlighted the large influence of late season versus early season burns (Figure 7a). Compared with no fire, early season biennial fires significantly reduced canopy below 5.5 m and late season biennial fire reduced canopy even further up to the 9.5 m height class ($p < 0.05$, Figure 7b), generating a vertical profile similar in shape but with lower frequency of occurrence. The late season fire profile contained significantly less canopy in nearly all height classes compared to the unburnt ($p < 0.05$, Figures 7a,b), but the most marked effects were in the lower height classes which represent the shrub layer and the recruitment zone.

## 4 Discussion

Airborne LiDAR provided direct measures of canopy cover and height distribution, and the derived metrics successfully predicted field-based estimates of aboveground biomass. The synoptic view that airborne LiDAR provided enabled us to map changes in biomass under different fire regimes, in addition to exploring differences in vegetation vertical profiles across the full expanse of the fire experiment.

### 4.1 Carbon storage consequences of altered fire regimes

Ten years of experimental burning imparted large structural differences in woody canopy across the plots of the Territory Wildlife Park fire experiment. Fire effects were most pronounced at the extremes of the experimental spectrum, with highest cover and biomass occurring under complete fire exclusion and lowest values of woody canopy structure obtained under biennial late season burning. The directionality of these trends was persistent across the underlying gradient of increasing soil depth and moisture, but the magnitude and slope of the effects was greater in the A and B block with shallower, drier soils

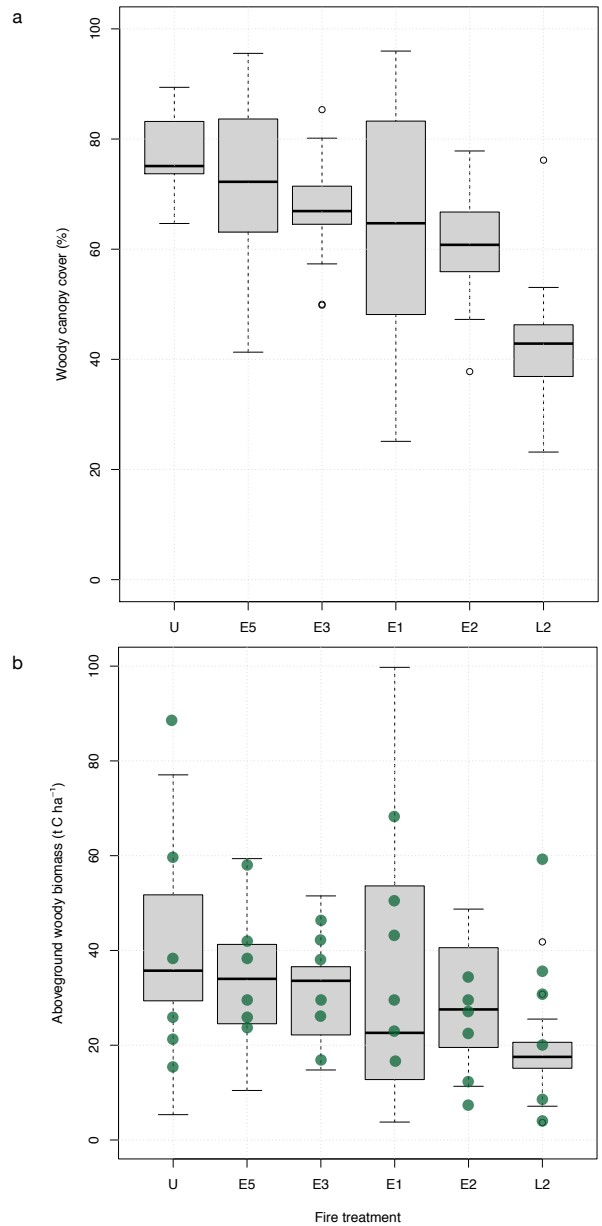

**Figure 4.** Relationship between fire treatments and (a) woody canopy cover and (b) woody biomass. Fire treatments ordered according to increasing fire intensity. Green dots in (b) indicate field measured values derived from 30 m X 30 m subplots.

(Figure 5). The lower magnitude of carbon reduction in the lower lying "C" block likely stems from the sparse herbaceous cover in these plots which results in patchy, low intensity fires.

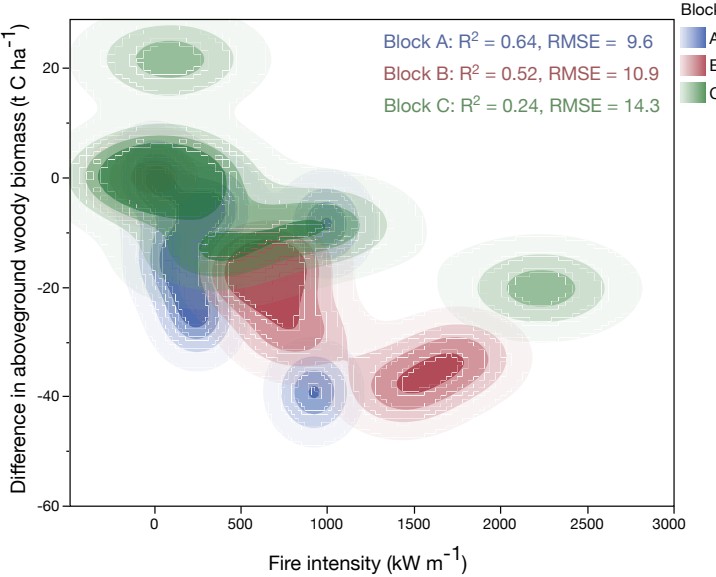

**Figure 5.** Density plot showing the relationship between increasing fire intensity reduction in woody carbon storage, relative to unburnt plots, for the A (blue), B (red) and C (green) blocks.

Recent research into woody biomass trends in the region (from long-term field monitoring plots) indicate that woody biomass has been relatively stable over decadal periods, with minor evidence of woody thickening, and that biomass is negatively correlated with fire frequency (Murphy et al., 2013). However, as Fensham et al. (2017) note, a key finding emerging from that regional study was that the observed decreases in tree biomass following severe fires were not driven by mortality of individual

trees, but rather by decreases in the rates of biomass accumulation of surviving trees. We do not have repeated individual tree data in our study to directly corroborate this finding, but the patterns of reduced cover throughout the height profile do suggest mortality and the consumption of trees by fire, rather than just reduction in growth rates.

Similar investigations in southern African savannas have found that fire frequency itself had little bearing on woody cover, but that the presence of fire alone was a stronger predictor of reduced woody cover (Devine et al., 2015). In our study however,

we found that cover and biomass were reduced as fire frequency increased (Figure 4), with the exception to the trend being the early biennial fires (E2), which had a slightly larger impact on structure than the early annual (E1) fires. The experimental design incorporates fire frequency and season, but the net result of these components of the fire regime is fire intensity, which is the stronger determining factor of vegetation structural change (Williams et al., 1999; Furley et al., 2008). Of all the fire treatments, the biennial burns had the highest mean intensities of 929 kW m$^{-1}$ and 1664 kW m$^{-1}$ for early season and late

season fire respectively. These intensities are still low compared to those of large late season fires in northern Australia, and reflect the small scale of the experimental plots. Nonetheless, despite lower intensities across the board compared to larger experiments like those obtained at the Kapalga experiment, our finding are in agreement with the diminished basal areas observed there under very high intensity late season fires (Andersen et al., 2003).

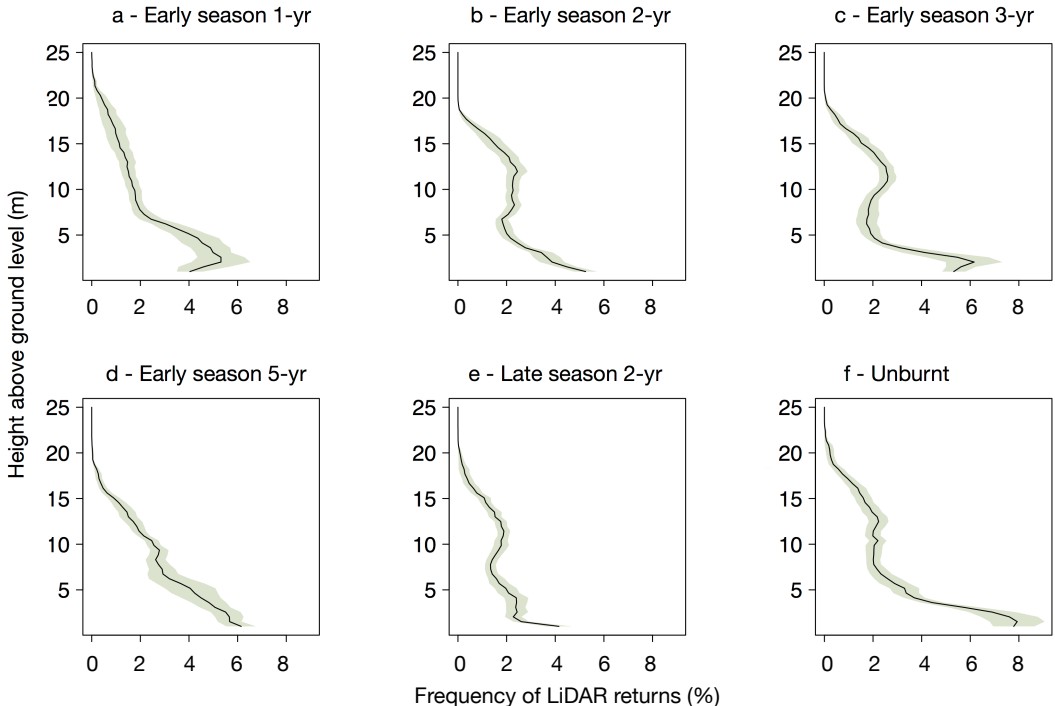

**Figure 6.** Effects of fire regime on vertical habitat structure determined from the frequency of airborne LiDAR returns. Solid black lines are the mean frequency distribution of LiDAR returns, and the green bands indicate the 95% confidence interval.

There is increasing interest in understanding the effect of different fire regimes on carbon stored in Australian savannas (Murphy et al., 2013; Cook et al., 2015) and recent studies have shown higher carbon stocks in dead organic matter under lower fire frequencies (Cook et al., 2016). At the Territory Wildlife Park fire plots the early biennial fire caused a reduction of 10 t C ha$^{-1}$ on average compared to unburnt plots, whereas late biennial fires almost doubled that average reduction to 19 t C ha$^{-1}$ (Figure 5). These patterns are consistent with the trend of lower greenhouse gas emissions under early dry season fires, relative to late fires (Meyer et al., 2012) and point to the importance of available fuel load and its characteristics (greater herbaceous volume and lower moisture content late in the dry season) in understanding fire induced structural change in savannas. This is further emphasised by the variation in response to fire along the environmental gradient of the experimental site.

Murphy et al. (2013) suggested that the moderation of fire regimes in northern Australia is likely to increase carbon storage in woody biomass, but the extent to which woody biomass can increase in these savannas is highly uncertain. Our results reduce some of this uncertainty, by providing quantification of the degree to which carbon stored in unburnt plots deviates from a range of different fire frequencies.

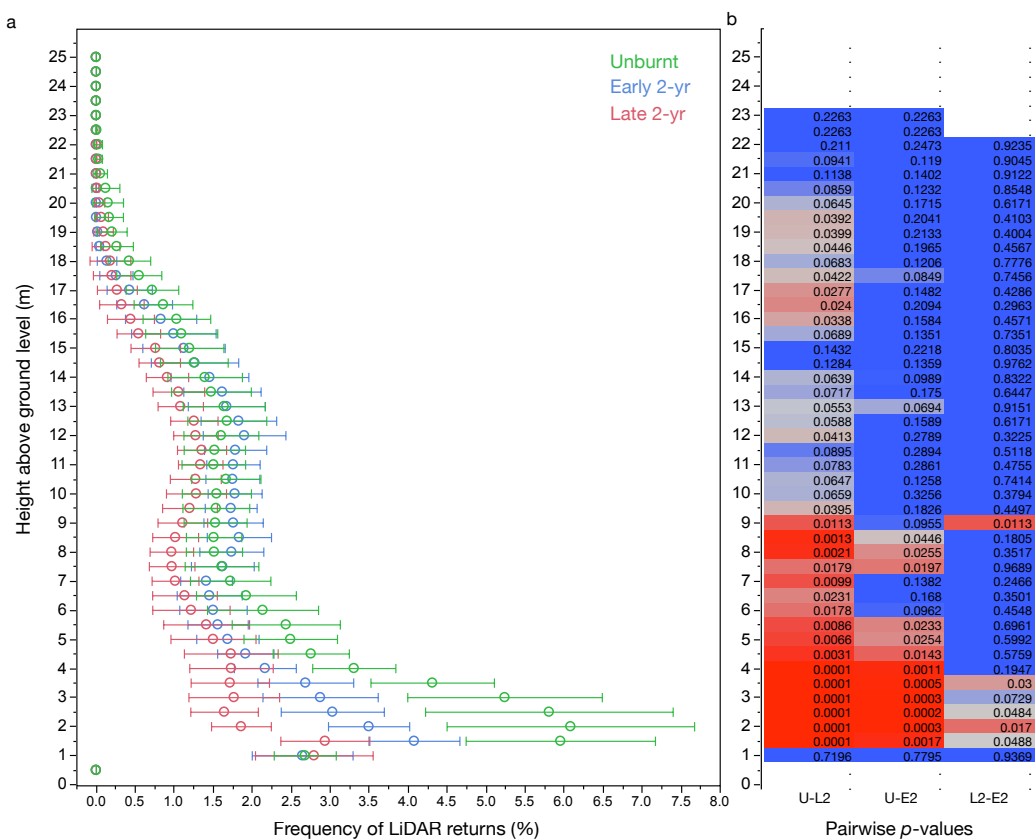

**Figure 7.** Effect of biennial fire season on woody vertical profile structure with unburnt treatment in green, early season biennial in blue, and late season biennial in red. (a) Dots and error bars represent mean and 95% confidence interval for the 30 m x 30 m subplots (n = 18). Some returns from lower height classes may have arisen from herbaceous material. (b) Pairwise comparison of differences in means on a per height class basis. U-L2 = unburnt vs late season biennial, U-E2 = unburnt vs early season biennial, L2-E2 = late season vs early season biennial. *p*-values are shaded with values < 0.05 in red.

## 4.2 Shifts in vegetation vertical profile distribution under altered fire regimes

Different fire regimes imparted a diverse array of vertical structural profiles on woody vegetation. Although woody canopy cover and aboveground biomass displayed subtle responses among the early season fire frequency treatments, we found that each fire regime generated a relatively unique niche space in terms of vertical profile distribution. These niches were most divergent in the understory height classes (< 5 m). Tracking these profiles over time into the future might reveal increased height of divergence as cohorts grow taller. Alternatively, these understorey height curves may represent stable persistent equilibrium resprout heights that define the optimum for resprouts that are able to persist within the flame zone under a particular fire regime (Freeman et al., 2017).

These vertical profile findings highlight the powerful role that fire management can play in shaping three-dimensional habitat structure in ecosystems. The challenge this presents to land-managers is deciding which of this range of profiles is optimal for their specific management objectives. We still lack explicit understanding of how different organisms utilize three-dimensional space, and it is increasingly evident that no one profile is optimal. Mid-story shrubs and trees provide key food resources for birds and small mammals, and high ground cover reduces predation risk by feral cats (Davies et al., 2016). Conversely, habitat simplification through late season burning was found to promote longer-term abundance of Frilled-neck lizards in Kakadu National Park, despite high initial direct mortality rates (Corbett et al., 2003; Andersen et al., 2005). As such, it is likely that a mix of patches at the landscape scale, spanning a diverse range of vertical profiles, is needed from a wildlife conservation perspective. The relative proportions and spatial arrangement of these patches needs targeted and deeper investigation.

## 4.3   Limitations and future directions

Our findings in this study provide quantification of the magnitudes of fire regime effects on woody structure in a tropical savanna. When generalizing to other savanna regions however, the following limitations should be to be taken into consideration. First, prior to the establishment of the TWP fire experiment in 2004 the vegetation was unburnt since 1990. Fourteen years of fire exclusion is rare in these tropical landscapes, so the starting conditions are atypical.

Second, despite the good results obtained in upscaling field-based woody biomass estimates with airborne LiDAR (Figure 3), future efforts should focus on reducing the level of uncertainty in the LiDAR-biomass model. Greater confidence in biomass/carbon prediction could be achieved by turning to individual tree-based segmentation approaches. Developments in terrestrial LiDAR in particular show great promise for providing individual tree volumes and biomass estimates that can be scaled, together with their uncertainties, to plot and landscape scales (Calders et al., 2014; Levick et al., 2016; Singh et al., 2018). Furthermore, the rich 3D models that terrestrial LiDAR provide will open up new avenues for exploring actual 3D structural metrics.

Last, our analyses in this study rely on differences between treatments at a single point in time to infer the mechanisms underpinning woody structural modification. Although typical for this type of investigation, the single time point approach should ideally be complimented with time-series analyses of before and after fire events to better constrain the mechanisms underpinning structural change.

## 5   Conclusions

We quantified the magnitude of aboveground carbon reduction under different regimes by integrating airborne LiDAR, field-surveys, and an ongoing fire regime experiment. Our results highlight the impact of late season burning on both carbon storage and on canopy vertical profile structure. Clear relationships between biodiversity and fire regimes have proven difficult to establish in savannas, despite many attempts at linking floral and faunal diversity directly to fire regime patterns. The range of vertical profile responses that we have illustrated here under different experimental fire treatments could hold the key to unlocking stronger links between fire management and biodiversity responses. High-resolution LiDAR can expose the structural

consequences of different management actions, and make them more easily accessible for integration with biodiversity and ecosystem process studies.

*Competing interests.* The authors declare no competing interests.

*Acknowledgements.* This project was jointly funded through CSIRO, Charles Darwin University, and the Australian Government's National
5 Environmental Science Programme (NESP). We acknowledge the Northern Territory Government and the Territory Wildlife Park for establishing and maintaining the long-term fire experimental site. Jorg Hacker and Airborne Research Australia (ARA) are thanked for their airborne surveying of the site. SRL was supported by grant number 01DR14010 from the BMBF (FIREBIODIV project).

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
