# Peer review of "Rapid response of habitat structure and aboveground carbon storage to altered fire regimes in tropical savanna"

_Biogeosciences, 2018_

## Referee Comment (RC1) · Anonymous Referee #1 · 29 Jun 2018

– General comments –

This manuscript reports the effects of various fire treatments, including combinations of early/late season burning and variation in fire frequency over more than a decade, on aboveground vegetation biomass and structure in a savanna habitat in northern Australia as quantified by airborne lidar. The results are relatively straight-forward and offer quantitative data that may be useful in future models of carbon dynamics and management of vegetation structure or heterogeneity. There is an interesting spatial interaction effect in which the results of a fire treatment depend on soil moisture and soil depth; the paper would be improved if these results were further explored and

elaborated upon. I would have appreciated seeing a validation of their model results on out-of-sample data to assess the accuracy of their results and the significance of the soil depth/moisture gradient. Some of the conclusions with respect to what is driving the observed decreases in woody cover with increasing fire intensity (i.e., greater tree mortality or reduced accumulation of woody biomass) appear to be unsubstantiated and require further explanation. Overall the paper, while not especially novel, does represent an important contribution to the literature by quantifying the effects of various fire regimes on 3-dimensional structure and aboveground biomass in northern Australian savannas.

– Specific comments –

Abstract: In the abstract there are inconsistent statements about the temporal scale of the experiment and how to interpret the results with respect to time. On page 1, line 3 the experiment is referred to as 'long-term', whereas on page 1, line 12 the results as said to have occurred over time scales as 'short' as a decade. It is important that the authors represent a consistent message: in their expert opinion, do structural changes occurring over a decade represent short-term or long-term responses? The title suggests that the interpretation is one that these are rapid changes and therefore observing these plots over ten years is not a particularly long time in the savanna tree cover cycle.

Page 2, lines 11-22: While declines in faunal populations are certainly important, I was surprised by the one-sided discussion of negative effects of savanna fires (e.g., the effects of savanna fires on greenhouse gas emissions). I felt this section of the manuscript lacked a balanced discussion of fire as an evolutionary force in savannas that, when suppressed, can have negative effects on savanna flora and fauna. True that some faunal populations are influenced but what about savanna specialists or species that rely on grass cover? Are there no species in these savannas that benefit from fire? Given the global and historical significance of fire in savannas, I advocate for a more balanced discussion of fire as a natural part of savanna landscapes that, when

well-managed, can have beneficial effects.

Page 2, lines 28-29: Is the significance here only that the approach is novel for savannas? Because lidar has been used to study fire effects in many other systems. Also, why is Smit et al. 2010 and your 2009 paper (Levick et al. 2009) not credited with studying fire effects on savanna vegetation structure using lidar? The Smit et al. 2010 paper was squarely aimed at "...assessing vegetation biomass and structural diversity responses to experimental fires"

Page 2, line 34: aim 1 is somewhat weak considering that lidar has been used successfully to study vegetation biomass and structure in so many other systems. It seems that we already know the answer to the question about reliably detecting vegetation and biomass and structure by airborne lidar is 'yes'. This first aim also puts the emphasis of the paper on methodology and thresholds of detection, which, in my opinion, changes the nature of the paper and requires more of a methodological approach. My suggestion is to leave this part out of future versions and focus on the effects of fire in this system.

Page 3, Table 1: This table legend is incomplete – are these mean fire intensity values? Also, I suggest you include standard errors or ranges for the fire intensity values (i.e., range for E5 and +/- SE for others).

Page 4, eqn (1); is there a different equation for multi-stemmed shrubs? Are they a significant part of the carbon pool?

Page 6, lines 8-10: this seems like a very comprehensive model which fits the data well (e.g., Fig. 3), but I am worried that there was no validation on out-of-sample data, which is the gold standard of model assessment. Perhaps it is challenging due to the paucity of lidar data, but is there any capacity to validate the model on out-of-sample data to get a better sense of model accuracy? It will also provide a means to understand the generality of eqn (2) to represent aboveground woody biomass with lidar derived data from this study (versus having to derive a new eqn for woody biomass at a different

site).

Page 7 and results section throughout: I strongly advise that when values are being reported, such as 75% or 45% canopy cover, the authors include some reasonable representation of error or variation (be it standard error or standard deviation, doesn't matter).

Page 7, Fig. 3 legend: text is incomplete. One should be able to look at the figure and legend and understand what information is being conveyed. This figure legend leaves much to be desired (location, sample size, where the data came from, refence to the model, etc.).

Page 8, lines 1-4: I found the fire * block interaction to be very interesting and worthy of some further exploration or analysis. I think your audience would be interested to know more about this interaction – are there other ancillary data that could help you explore this soil/moisture effect? To begin with, the directionality of the interaction is never reported – does greater depth/moisture increase or decrease the effect of a given fire treatment on woody cover and biomass? At the very least this should be reported. Further, once the directionality is presented, what is the mechanistic nature of this interaction? Is it related to quantity or composition of the fuel as depth and soil water availability changes? This question would be helped by data if you have it, otherwise perhaps a few sentences in the discussion are in order.

Page 8, lines 22-24: like my comment above, I did not find this conclusion or aim very compelling since we already know these methods work well and this is not a methods paper. I recommend sticking to the ecological effects of fire in these tropical savannas as the main focus of the paper.

Page 8, line 30: is this interpretation entirely correct? Wasn't there an interaction effect between fire treatment and block suggesting that the fire treatments did not simply 'persist' but in fact 'changed' with soils moisture and depth (i.e., the interaction effect). I suggest a re-evaluation of this simple interpretation and better presentation of what

are interesting interaction effects.

Page 9, lines 2-3 and page 10, lines 1-3: I do not understand how this conclusion (that decreasing biomass was the result of decreasing biomass accumulation rather than mortality) was reached from this study. The text and the citation of Fensham et al. 2017 suggests that the result and conclusion come from another study rather than this one – is that the case? Moreover, the statement on page 10 is confusing because it suggests that your interpretation of the data is that mortality from fire is a driving factor in the observed patterns (in direct contracts to the sentence on page 9). Either way, clarification and rewriting are required here, as we don't know where these conclusions are coming from and there is no evidence that the current study can provide demographic data of the nature being described here.

Page 10 & 11: If my interpretation is correct, Figs 6 and 7 are representing the same data. Consequently, it may make more sense to represent Fig. 7 as a difference from the control plot rather than as the same data presented in Fig. 6 (would that make sense?).

– Technical corrections –

Page 7, Table 2: delta AIC for the top model should be reported as 0.00.

Page 8, line 27: should read "...in woody canopy cover..." or "...in woody canopy structure..."

Page 10, Fig. 5 legend: should the legend read: "Correlation between change in fire intensity and difference in woody canopy cover..."? Also, it needs to be clear what is meant by change in fire intensity; is this control – treatment or some other metric. More text and greater clarity (which is the case with almost all the figure legends in this paper).

---

## Referee Comment (RC2) · Anonymous Referee #2 · 3 Jul 2018

Levick et al. Rapid response of habitat structure and aboveground carbon storage to altered fire regimes in tropical savanna.

This is a useful application of LiDAR technology to examine effects of burning on vegetation structure. The results are important, but I must admit that I was disappointed there were no analyses of how fire affected 3D vegetation structure, despite multiple claims to the contrary (Page 1, lines 8 and 11; Page 2, line 34; Page 12, Line 13; Page12, line 17 Figure 6, caption). These claims should be removed or actual analysis of 3D structure should be added. Figure 2 is a great reconstruction of the 3D structure of the vegetation, but the information contained therein was ultimately distilled into met-

rics that lose this 3D information. I do not have the expertise to suggest what metrics should be used to compare 3D structure, but certainly such metrics must exist, such as the various methods to measure aggregation. It would have been helpful to have a brief overview of the research approach at the end of the introduction. For example, as I was reading the methods, it was not clear to me why you used Lidar to estimate biomass of the fire plots when you already had more direct measurements of above-ground biomass for the same plots. Of course your approach allowed you to estimate biomass for a 3-fold greater area of each experimental plot, which I suspect is the reason that you did this, but this was not clearly laid out. Considering that you possess the ground-based data for comparing fire impact on AGB, a direct test using these data should be included. Even though the area sampled is lower, the ground measurements avoid the additional error introduced by relying on a model relationship (even though the fit was quite good). What is the difference between Figure 7 and the corresponding data from figure 6? At first glance, it appeared that Figure 7 was presenting data already presented in figure 6, but upon close examination, the corresponding data in figure 6 are different than figure 7. For example in figure 6, there is more vegetation at heights of about 8 to 15m in the 2-yr early treatment than in the unburnt treatment, in contrast to Figure 7. The figure legends and text do not help clarify these differences. Also, are the error bars standard errors? Were they calculated using variation and n of 30x30 plots or of experimental plots? The latter should be used if we are to use them to compare treatments. The fire intensity data in Table 1 are important for this study, but no details are given. How were these data collected? Were they obtained for every fire between 2004 and 2013 or just for representative fires? If these data have not been published elsewhere then the methods should be described. Page 2, Line 23. It seems like an overstatement that detailed 3D measurements are the best way to quantify carbon dynamics. Perhaps it could be the best choice for non-destructive measurements of certain C pools. Page 3, line 15 and line 19. In these instances replace "blocks" with "block." Page 4, line 3. In what year were these tree measurements made? Page 5, lines 8-12 and page 6, line 3. Are references available for these software tools? Page 6, line 12. I presume that two of these six quadrats corresponded with the plots sampled on the ground. It would be helpful to clarify this. If not, I am not sure how figure 3 was generated. Page 6, line 15. I disagree that including quadrats as a random resolves the issue of pseudoreplication. One foolproof way of avoiding pseudoreplication would be to average your data across quadrats to get a single value for each experimental plot. Traditionally the blocks are considered to provide the replication, but this is lost if block and block x treatment are treated as a fixed factors. For a randomized full block design, block is typically treated as a random factor, treating the blocks as replicates of the experimental treatment, and in a least-squares approach, the block x treatment interaction would be used for the denominator MS. Of course the denominator df would be rather small in a design like this. I am not quite sure what is accomplished by treating the subplot as a random factor, but certainly it is not eliminating the pseudoreplication issue. I believe there are ways of estimating df for lme4 tests, and these should be presented, and I strongly recommend that the authors archive their data and r code as supplementary information. All this being said, this is a large-scale experiment, which commonly suffer from pseudoreplication, so I am not as concerned about pseudoreplication here as I am about the claim that pseudoreplication has been avoided. Figure 3. The legend should state what each point represents. I presume the ground-estimated AGB corresponds to one 30m x 30m plot. Page 7, line 6-7. I don't think is what you really mean to say. It is always true that the model including all factors and interactions will explain the most variance. Besides, Table 2 doesn't really show how much variance is explained. Page 8, line 18. It is stated here that the late burns had significantly less canopy than the unburnt, but no statistical tests were performed. Perhaps this conclusion is based on the non-overlap of error bars in figure 7. This should be clarified, and it is important to provide details on how these errors bars were generated. Page 9, Line 2. It isn't clear what "this study" is. Does it refer to the present study, to Murphy et al 2013, or to Fensham et al 2017? Figure 5. Are these relationships significant if you do not aggregate them by treatment? Presumably you have fire intensity data for each 1-ha plot, which would allow you to test this for a larger

number of true replicates. Page 10, Lines 1-3. Please be specific about what results from your study suggest this. Figure 6. Please provide more information about the data in this figure. Are these frequency distributions of the returns themselves, or are they a reconstruction of vegetation density that takes into account the fact that foliage high in the canopy has a higher probability of being detected than foliage low in the canopy. Also, figure 6 shows 1-D vegetation structure, not 3-D structure as indicated by the caption. Page 11, Line 3. Where do you show this correlation? You show a relationship with fire intensity, but I don't think you showed this for frequency. Page 12, line 3. This mention of herbaceous volume here raises a relevant point regarding the interpretation of your figures. In figure 7, do the data corresponding to 1-m above the ground correspond in reality to 0-1 m, or to 1-2 m, or to 0.5 to 1.5 m. When looking at figure 7, it wasn't clear whether grasses would be included in the lowest point. Page 12, line 12. I am not sure what minimal overlap means here. I don't think you are referring to overlap of individual trees, since you did not examine this. And looking at figure six, I would say that there is a lot of overlap in these distributions, since some distributions fit wholly within others.

---

## Author Comment (AC1) · 9 Aug 2018

Reviewer comment: This manuscript reports the effects of various fire treatments, including combinations of early/late season burning and variation in fire frequency over more than a decade, on aboveground vegetation biomass and structure in a savanna habitat in northern Australia as quantified by airborne lidar. The results are relatively straight-forward and offer quantitative data that may be useful in future models of carbon dynamics and management of vegetation structure or heterogeneity.

Author response: Thank you, we do hope that this work proves useful to the modelling community.

Reviewer comment: There is an interesting spatial interaction effect in which the results of a fire treatment depend on soil moisture and soil depth; the paper would be improved if these results were further explored and elaborated upon. I would have appreciated seeing a validation of their model results on out-of-sample data to assess the accuracy of their results and the significance of the soil depth/moisture gradient.

Author response: We agree with the points you have raised and have addressed them in detail in the specific comments section below.

Reviewer comment: Some of the conclusions with respect to what is driving the observed decreases in woody cover with increasing fire intensity (i.e., greater tree mortality or reduced accumulation of woody biomass) appear to be unsubstantiated and require further explanation.

Author response: We have modified the sentences in question for clarification.

Reviewer comment: Overall the paper, while not especially novel, does represent an important contribution to the literature by quantifying the effects of various fire regimes on 3-dimensional structure and aboveground biomass in northern Australian savannas.

Author response: Thank you for your comments, they have helped shape a much stronger manuscript.

Reviewer comment:: In the abstract there are inconsistent statements about the temporal scale of the experiment and how to interpret the results with respect to time. On page 1, line 3 the experiment is referred to as 'long-term', whereas on page 1, line 12 the results as said to have occurred over time scales as 'short' as a decade. It is important that the authors represent a consistent message: in their expert opinion, do structural changes occurring over a decade represent short-term or long-term responses? The title suggests that the interpretation is one that these are rapid changes and therefore observing these plots over ten years is not a particularly long time in the savanna tree cover cycle.

Author response: Very true, thanks for highlighting this. We tend to think of the experiment as a long-term one as we plan to maintain it for decades to come. However we agree that the current timespan of the experiment is short in context of savanna tree cover cycles, which is why we were impressed with the degree of change that has occurred and used 'rapid' in the title. We have made changes to our terminology throughout to avoid this ambiguity and no longer refer to it as a long-term experiment.

Reviewer comment: Page 2, lines 11-22: While declines in faunal populations are certainly important, I was surprised by the one-sided discussion of negative effects of savanna fires (e.g., the effects of savanna fires on greenhouse gas emissions). I felt this section of the manuscript lacked a balanced discussion of fire as an evolutionary force in savannas that, when suppressed, can have negative effects on savanna flora and fauna. True that some faunal populations are influenced but what about savanna specialists or species that rely on grass cover? Are there no species in these savannas that benefit from fire? Given the global and historical significance of fire in savannas, I advocate for a more balanced discussion of fire as a natural part of savanna landscapes that, when well-managed, can have beneficial effects.

Author response: Good point. We have restructured this section as suggested and have now presented a more balanced perspective, including the importance of fire in savanna ecosystem functioning.

Reviewer comment: Page 2, lines 28-29: Is the significance here only that the approach is novel for savannas? Because lidar has been used to study fire effects in many other systems. Also, why is Smit et al. 2010 and your 2009 paper (Levick et al. 2009) not credited with studying fire effects on savanna vegetation structure using lidar? The Smit et al. 2010 paper was squarely aimed at ". . .assessing vegetation biomass and structural diversity responses to experimental fires"

Author response: We have rephrased this section. Although the Smit et al 2010 and Levick et al 2009 papers are relevant here in that they utilised airborne LiDAR across
fire experiments in savanna, but neither of paper quantified biomass and its variation across fire treatments (only height).

Reviewer comment: Page 2, line 34: aim 1 is somewhat weak considering that lidar has been used successfully to study vegetation biomass and structure in so many other systems. It seems that we already know the answer to the question about reliably detecting vegetation and biomass and structure by airborne lidar is 'yes'. This first aim also puts the emphasis of the paper on methodology and thresholds of detection, which, in my opinion, changes the nature of the paper and requires more of a methodological approach. My suggestion is to leave this part out of future versions and focus on the effects of fire in this system.

Author response: Thanks you - valid comment. The goal here was indeed to focus on the fire effects, so we have restructured the aims to focus more squarely on the ecology. We agree that the answer to reliable detection of vegetation structure by LiDAR is "yes" – but what needs deeper consideration is the sensitivity of these techniques to detecting change. In our case, is the degree of structural change caused by fire manipulation greater than the uncertainty associated with LiDAR biomass estimation?

Reviewer comment: Page 3, Table 1: This table legend is incomplete – are these mean fire intensity values? Also, I suggest you include standard errors or ranges for the fire intensity values (i.e., range for E5 and +/- SE for others).

Author response: Updated as suggested, with SE included.

Reviewer comment: Page 4, eqn (1); is there a different equation for multi-stemmed shrubs? Are they a significant part of the carbon pool?

Author response: Very good question. Shrubs are generally ignored, and are considered to be a minor part of the carbon pool. However they are an important part of the ecosystem and some represent future trees. We have not accounted for shrubs well in either our fieldwork or our airborne LiDAR. We have now made this clear in the

manuscript and have added it to our limitations section. As a side note we have started new projects exploring the shrub component with ground-based LiDAR.

Reviewer comment: Page 6, lines 8-10: this seems like a very comprehensive model which fits the data well (e.g., Fig. 3), but I am worried that there was no validation on out-of-sample data, which is the gold standard of model assessment. Perhaps it is challenging due to the paucity of lidar data, but is there any capacity to validate the model on out-of-sample data to get a better sense of model accuracy? It will also provide a means to understand the generality of eqn (2) to represent aboveground woody biomass with lidar derived data from this study (versus having to derive a new eqn for woody biomass at a different site).

Author response: We agree that out-of-sample data would be ideal for further independent validation. Unfortunately this is not possible with the data we have, and with the time that has passed since the LiDAR flight was conducted. Despite this, we have added our field estimated C values to Figure 4b so that the value for each 30 m X 30 m plot is now overlaid on top of the LiDAR derived values in the box plots. A key point here is that interpretation of biomass changes across the fire treatments does not differ if using the original field data or the LiDAR derived model – providing greater confidence in the ecological conclusions we are drawing.

Reviewer comment: Page 7 and results section throughout: I strongly advise that when values are being reported, such as 75% or 45% canopy cover, the authors include some reasonable representation of error or variation (be it standard error or standard deviation, doesn't matter).

Author response: Updated as suggested.

Reviewer comment: Page 7, Fig. 3 legend: text is incomplete. One should be able to look at the figure and legend and understand what information is being conveyed. This figure legend leaves much to be desired (location, sample size, where the data came from, refence to the model, etc.).

Author response: Agreed – updated accordingly, and figure legends improved through-out.

Reviewer comment: Page 8, lines 1-4: I found the fire * block interaction to be very interesting and worthy of some further exploration or analysis. I think your audience would be interested to know more about this interaction – are there other ancillary data that could help you explore this soil/moisture effect? To begin with, the directionality of the interaction is never reported – does greater depth/moisture increase or decrease the effect of a given fire treatment on woody cover and biomass? At the very least this should be reported. Further, once the directionality is presented, what is the mechanistic nature of this interaction? Is it related to quantity or composition of the fuel as depth and soil water availability changes? This question would be helped by data if you have it, otherwise perhaps a few sentences in the discussion are in order.

Author response: Very good point – we have expanded on this interaction and have made the directionality clear. We have also expanded on proposed mechanisms in the Discussion.

Reviewer comment: Page 8, lines 22-24: like my comment above, I did not find this conclusion or aim very compelling since we already know these methods work well and this is not a methods paper. I recommend sticking to the ecological effects of fire in these tropical savannas as the main focus of the paper.

Author response: Agreed – we have restructured the aims to focus squarely on the ecological effects.

Reviewer comment: Page 8, line 30: is this interpretation entirely correct? Wasn't there an interaction effect between fire treatment and block suggesting that the fire treatments did not simply 'persist' but in fact 'changed' with soils moisture and depth (i.e., the interaction effect). I suggest a re-evaluation of this simple interpretation and better presentation of what are interesting interaction effects.

Author response: Thanks for picking up this point – agreed and modified as suggested.

Reviewer comment: Page 9, lines 2-3 and page 10, lines 1-3: I do not understand how this conclusion (that decreasing biomass was the result of decreasing biomass accumulation rather than mortality) was reached from this study. The text and the citation of Fensham et al. 2017 suggests that the result and conclusion come from another study rather than this one – is that the case? Moreover, the statement on page 10 is confusing because it suggests that your interpretation of the data is that mortality from fire is a driving factor in the observed patterns (in direct contracts to the sentence on page 9). Either way, clarification and rewriting are required here, as we don't know where these conclusions are coming from and there is no evidence that the current study can provide demographic data of the nature being described here.

Author response: It was not our intention to suggest that decreasing biomass accumulation rather than mortality was the driver. In hindsight we can see how it could have been read like this and have modified this section to avoid any confusion. Likewise we have rewritten these sentences to remove ambiguity between interpretations from our study and the literature referenced.

Reviewer comment: Page 10 & 11: If my interpretation is correct, Figs 6 and 7 are representing the same data. Consequently, it may make more sense to represent Fig. 7 as a difference from the control plot rather than as the same data presented in Fig. 6 (would that make sense?).

Author response: They are similar although Figure 6 showed mean vertical profile and 95% CI for all treatments, while Figure 7 shows only unburnt, 2 year early season and 2 year late season with SE of the mean. We have tried the suggestion of plotting Figure 7 as the difference to the unburnt, however we consider the direct comparison of the unburnt condition and the different season burns to be valuable. We have expanded and clarified the figure legends.

Reviewer comment: Page 7, Table 2: delta AIC for the top model should be reported

as 0.00.

Author response: Thank you – corrected.

Reviewer comment: Page 8, line 27: should read "...in woody canopy cover..." or "...in woody canopy structure. . ."

Author response: Fixed.

Reviewer comment: Page 10, Fig. 5 legend: should the legend read: "Correlation between change in fire intensity and difference in woody canopy cover. . ."? Also, it needs to be clear what is meant by change in fire intensity; is this control – treatment or some other metric. More text and greater clarity (which is the case with almost all the figure legends in this paper).

Author response: Thank you, we have clarified this legend and have been through all the other legends to provide more detail.

---

## Author Comment (AC2) · 9 Aug 2018

Reviewer comment: This is a useful application of LiDAR technology to examine effects of burning on vegetation structure. The results are important, but I must admit that I was disappointed there were no analyses of how fire affected 3D vegetation structure, despite multiple claims to the contrary (Page 1, lines 8 and 11; Page 2, line 34; Page 12, Line 13; Page12, line 17 Figure 6, caption). These claims should be removed or actual analysis of 3D structure should be added.

Author response: Thank you, we're glad you consider these results to be important. Our reference to 3D comes from our consideration of canopy cover (horizontal component) and height (vertical component) which together encompass the 3D structure of vegetation. However we agree with your comment that we have not analysed single metrics that capture 3D structure/diversity. We have removed any misleading claims and have checked the validity of our terminology throughout.

Reviewer comment: Figure 2 is a great reconstruction of the 3D structure of the vegetation, but the information contained therein was ultimately distilled into metrics that lose this 3D information. I do not have the expertise to suggest what metrics should be used to compare 3D structure, but certainly such metrics must exist, such as the various methods to measure aggregation.

Author response: Thanks you. We disagree that the 3D information has been lost through our analyses, it has been distilled and we focused on metrics which are targeted in traditional ecology (height, height layering, cover, biomass). The field of true 3D metrics is gaining momentum and we agree more could be done to derive metrics of full 3D structure. We consider this avenue to be important for future research, but beyond the scope of this study. We have raised this point in the future directions section of our discussion.

Reviewer comment: It would have been helpful to have a brief overview of the research approach at the end of the introduction. For example, as I was reading the methods, it was not clear to me why you used Lidar to estimate biomass of the fire plots when you already had more direct measurements of above- ground biomass for the same plots. Of course your approach allowed you to estimate biomass for a 3-fold greater area of each experimental plot, which I suspect is the reason that you did this, but this was not clearly laid out.

Author response: Thanks for pointing that out, and yes our reasoning to use LiDAR was to increase the area sampled, but also to test the potential for LiDAR to be used in future fire/biomass studies over much larger areas in these landscapes. We have laid this out more clearly at the end of the introduction.

Reviewer comment: Considering that you possess the ground-based data for comparing fire impact on AGB, a direct test using these data should be included. Even though the area sampled is lower, the ground measurements avoid the additional error introduced by relying on a model relationship (even though the fit was quite good).

Author response: Good point - we have added the field estimated AGB values to Figure 4b, making it possible to compare the patterns as if we only had field data available.

Reviewer comment: What is the difference between Figure 7 and the corresponding data from figure 6? At first glance, it appeared that Figure 7 was presenting data already presented in figure 6, but upon close examination, the corresponding data in figure 6 are different than figure 7. For example in figure 6, there is more vegetation at heights of about 8 to 15m in the 2-yr early treatment than in the unburnt treatment, in contrast to Figure 7. The figure legends and text do not help clarify these differences. Also, are the error bars standard errors? Were they calculated using variation and n of 30x30 plots or of experimental plots? The latter should be used if we are to use them to compare treatments.

Author response: It is the same underlying data. We have tried various iterations of showing all the profiles together, but found them too clustered for comparison. Figure 6 shows the vertical profile means and 95% CI. We broke out the unburnt and the early and late season 2-years to show the effect of altering only season while keeping only frequency constant, since early versus late season burning is important from a policy perspective in northern Australia. We also show the mean and SE (experimental plots) here to be more objective is comparing the overlap between treatments. Clarified in text and legends.

Reviewer comment: The fire intensity data in Table 1 are important for this study, but no details are given. How were these data collected? Were they obtained for every fire between 2004 and 2013 or just for representative fires? If these data have not been published elsewhere then the methods should be described.

[Figure]

Author response: We have added this to the methods section and provided a reference to earlier work.

Reviewer comment: Page 2, Line 23. It seems like an overstatement that detailed 3D measurements are the best way to quantify carbon dynamics. Perhaps it could be the best choice for non-destructive measurements of certain C pools.

Author response: True – modified to say that better understanding of above ground biomass can be achieved

Reviewer comment: Page 3, line 15 and line 19. In these instances replace "blocks" with "block."

Author response: Corrected.

Reviewer comment: Page 4, line 3. In what year were these tree measurements made?

Author response: 2014 – now specified in manuscript

Reviewer comment: Page 5, lines 8-12 and page 6, line 3. Are references available for these software tools?

Author response: Yes – now provided. rapidlasso GmbH, "LAStools - efficient LiDAR processing software", obtained from http://rapidlasso.com/LAStools

Reviewer comment: Page 6, line 12. I presume that two of these six quadrats corresponded with the plots sampled on the ground. It would be helpful to clarify this. If not, I am not sure how figure 3 was generated.

Author response: Correct – clarified.

Reviewer comment: Page 6, line 15. I disagree that including quadrats as a random resolves the issue of pseudoreplication. One foolproof way of avoiding pseudoreplication would be to average your data across quadrats to get a single value for each experimental plot. Traditionally the blocks are considered to provide the replication,

but this is lost if block and block x treatment are treated as a fixed factors. For a randomized full block design, block is typically treated as a random factor, treating the blocks as replicates of the experimental treatment, and in a least-squares approach, the block x treatment interaction would be used for the denominator MS. Of course the denominator df would be rather small in a design like this. I am not quite sure what is accomplished by treating the subplot as a random factor, but certainly it is not eliminating the pseudoreplication issue. I believe there are ways of estimating df for lme4 tests, and these should be presented, and I strongly recommend that the authors archive their data and r code as supplementary information. All this being said, this is a large-scale experiment, which commonly suffer from pseudoreplication, so I am not as concerned about pseudoreplication here as I am about the claim that pseudoreplication has been avoided.

Author response: Thanks for raising these concerns. We have removed claims that our approach has avoided pseudoreplication.

Reviewer comment: Figure 3. The legend should state what each point represents. I presume the ground-estimated AGB corresponds to one 30m x 30m plot.

Author response: Yes that's correct. We have updated this legend (and others) with more detail.

Reviewer comment: Page 7, line 6-7. I don't think is what you really mean to say. It is always true that the model including all factors and interactions will explain the most variance. Besides, Table 2 doesn't really show how much variance is explained.

Author response: Thanks for picking this up – we have clarified the text.

Reviewer comment: Page 8, line 18. It is stated here that the late burns had significantly less canopy than the unburnt, but no statistical tests were performed. Perhaps this conclusion is based on the non-overlap of error bars in figure 7. This should be clarified, and it is important to provide details on how these errors bars were generated.

Author response: Clarified as suggested, and details on error bars suggested.

Reviewer comment: Page 9, Line 2. It isn't clear what "this study" is. Does it refer to the present study, to Murphy et al 2013, or to Fensham et al 2017?

Author response: Thanks – we have clarified this section..

Reviewer comment: Figure 5. Are these relationships significant if you do not aggregate them by treatment? Presumably you have fire intensity data for each 1-ha plot, which would allow you to test this for a larger number of true replicates.

Author response: Good point – we have explored this in more detail and have used the non-aggregated intensity data. The refreshed Figure 5 now also shows the differences with landscape position as raised by Reviewer 1 (A,B,C block).

Reviewer comment: Reviewer comment: Page 10, Lines 1-3. Please be specific about what results from your study suggest this.

Author response: Clarified as requested.

Reviewer comment: Figure 6. Please provide more information about the data in this figure. Are these frequency distributions of the returns themselves, or are they a reconstruction of vegetation density that takes into account the fact that foliage high in the canopy has a higher probability of being detected than foliage low in the canopy. Also, figure 6 shows 1-D vegetation structure, not 3-D structure as indicated by the caption.

Author response: We have provided clearer information as requested. These are the returns after running a voxel thinning to remove duplicate points and standardise density across the site. Probability of upper layer detection is not explicitly accounted for - these effects are minimal in savannas compared to denser tropical or temperate systems. These details have been added and the Figure legend has been corrected.

Reviewer comment: Page 11, Line 3. Where do you show this correlation? You show a relationship with fire intensity, but I don't think you showed this for frequency.

Author response: True – this sentence has been revised.

Reviewer comment: Page 12, line 3. This mention of herbaceous volume here raises a relevant point regarding the interpretation of your figures. In figure 7, do the data corresponding to 1-m above the ground correspond in reality to 0-1 m, or to 1-2 m, or to 0.5 to 1.5 m. When looking at figure 7, it wasn't clear whether grasses would be included in the lowest point.

Author response: 1-m corresponds to 1-2m, we have clarified this in the Figure legend. Denser patches of grass may be included in the lower layers, but most often it is not detected. We have added a line stating there might be some returns coming from herbaceous layer, but we cannot quantify this.

Reviewer comment: Page 12, line 12. I am not sure what minimal overlap means here. I don't think you are referring to overlap of individual trees, since you did not examine this. And looking at figure six, I would say that there is a lot of overlap in these distributions, since some distributions fit wholly within others.

Author response: Thanks for picking this up – we did mean the distributions, but overlap was the wrong term, we have clarified this sentence.

---

## Author Response (AR2)

Dear Editor

Thank you for you feedback.

Based on the minor suggestions made during the second round of review we have updated our manuscript accordingly.

We have responded to the reviewer comments point-by-point on the next page and have provided a marked-up version showing the changes we have made.

We have also gone through the whole manuscript again and corrected some minor typos.

RC1: The authors present a straightforward interpretation of their approach to quantifying savanna structure and biomass using ground-based and lidar observations. In my view they have made appropriate changes to the manuscript and largely addressed the concerns of prior reviewers. However, I think the discussion of the statistical approaches used and the reporting of the results of the analyses remain insufficient.

Agreed. We have provided clearer detail.

RC2: First, Table 1 presenting the model selection results appears to be incomplete since it does not include AIC values for canopy height. In addition, while the model selection criteria are reported, the models themselves are not. Since these are ANOVAs it would seem appropriate to report the f-statistics and be clear about what model was selected in the end and why (for example, is the "best" model really one where just the interaction term is included and not the associate main effects?). More clarity in the statistical approach is needed along with a more complete reporting of the statistical results.

Thanks for picking that up, we have added canopy height to Table 2 as suggested. As these are not classic ANOVAs but linear mixed models with nested random effects, we feel that the AIC selection approach and reporting is appropriate and in line with current ecological literature.

RC3: Second, it would appear that the vertical structure treatment comparisons were based on visual assessments of the degree of overlap of confidence intervals from Figure 6. Is this the case? Key statements in the results have unclear statistical support, for example: "Compared with no fire, early season biennial fires reduced cover across all heights, but especially below 7m, and late season biennial fire reduced cover even further throughout, generating a vertical profile similar in shape but with much lower frequency of occurrence (Figure 7). The late season fire profile contained significantly less canopy in all height classes compared to the unburnt (no overlap of error bars), but the most marked effects were in the lower height classes (shrub layer)." The authors should be clear about how these conclusions are drawn, specifically which statistical results are used for these inferences and justify the statistical approach. In my view, this is lacking in the current version of the manuscript.

Agreed. We have clarified our interpretation and have added a new panel to Figure 7 (Fig. 7b) which includes p-values for the pairwise comparison on a per height class basis. The reviewer was correct in that we were discussing non-overlap of the 95% CI bands – we now compliment this with paired t-test results. We have also updated Figure 7a to use 95% CI bars instead of SE as in the previous version for consistency.

0394Δ 8710Δ 0331 014B 2032'

[revised manuscript text omitted]